# INFUSER: INjecting Synthetic FailUre for Self-Correcting Embodied Agents

## ABSTRACT

Vision-Language Models (VLMs) have become a powerful foundation for embodied agents, which are typically fine-tuned on expert demonstrations of successful task completions. However, collecting expert demonstrations is prohibitively expensive, and additionally, training exclusively on these ideal trajectories leaves agents brittle and struggle to recover from inevitable errors. To address this issue, we introduce INFUSER, INjecting synthetic FailUre for Self-correcting Embodied agent. Our idea is to augment existing expert trajectories with automatically generated failure-and-recovery scenarios (i.e., no human cost), rather than collecting additional (costly) expert demonstrations. Specifically, we synthesize these data by injecting suboptimal actions into ground-truth paths, creating a diverse set of controlled failure scenarios. By fine-tuning on this augmented dataset, INFUSER learns to take corrective actions and recover from mistakes. Our experiments validate the effectiveness of INFUSER through comprehensive evaluations on benchmarks for embodied agents including EB-ALFRED and EB-Habitat; training the Qwen2.5-VL-7B model by augmenting with our synthetic failure-tolerant data improves its performance by 18.3% → 47.0% and 59.7% → 66.3% on EB-ALFRED and EB-Habitat, respectively, achieving state-of-the-art performance among open-source models and even surpassing Qwen2.5-VL-72B with 10× fewer parameters. These results demonstrate that learning to recover from failures through synthetic augmentation, rather than collecting additional expert demonstrations, is a cost-effective approach to building robust embodied agents.

## 1 INTRODUCTION

Vision-Language Models (VLMs; OpenAI, 2025; Wang et al., 2025b; Anthropic, 2025b; Team et al., 2025) have demonstrated remarkable success across a range of multimodal tasks, including visual question answering (Liu et al., 2024), image captioning (Li et al., 2023), image-text retrieval (Xiao et al., 2025), and zero-shot learning (Saha et al., 2024). A recent frontier in VLM research seeks to extend these capabilities beyond understanding static image-text pairs to more complex, sequential decision-making problems (Klissarov et al., 2025; Rocamonde et al., 2024). A prominent example of this direction is the development of *embodied agents* (Driess et al., 2023; Wu et al., 2025; Shi et al., 2025; Wang et al., 2025a), which must reason about their environment and generate a sequence of actions to achieve a goal. The predominant paradigm for training these agents involves fine-tuning a VLM on expert demonstrations of successful task completion (Wu et al., 2025; Shi et al., 2025).

Despite this progress, current agents trained via supervised fine-tuning suffer from several limitations. First, collecting the necessary expert demonstrations is prohibitively expensive and labor-intensive, posing a significant barrier to scalability. Second, by training exclusively on flawless expert demonstrations, agents are only exposed to on-distribution, ideal state-action trajectories. Consequently, the agent lacks the knowledge to take corrective actions and recover when an error inevitably occur. Our analysis of error persistence patterns reveals this fundamental limitation: even state-of-the-art models like Claude-3.7-Sonnet (Anthropic, 2025a), despite achieving the fewest total errors, demonstrate the poorest recovery capabilities with only 65.4% of errors followed by successful actions (see Table 6 for more detail). This recovery paradox—where models optimized for precision lack resilience—highlights the critical need for explicit training on failure scenarios.

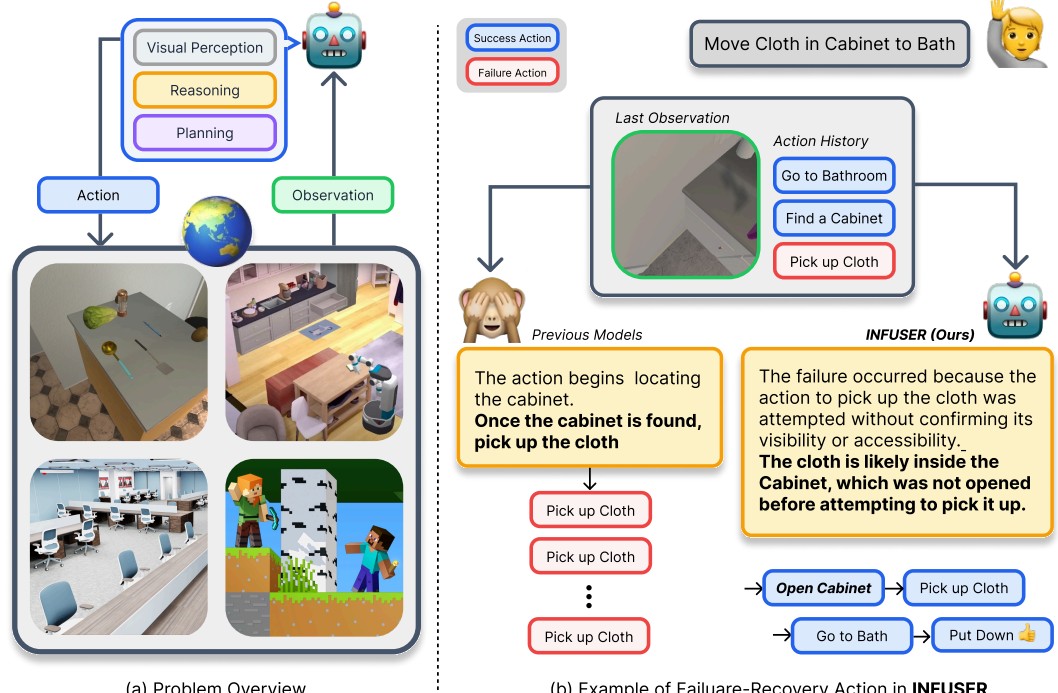

(a) Problem Overview      (b) Example of Failuare-Recovery Action in **INFUSER**

Figure 1: **INFUSER's problem overview and failure-recovery example.** (a) INFUSER enhances embodied agents to interact with environments to perceive visual scenes, reason about task states, and plan appropriate actions even after failures occur. Environments where INFUSER is evaluated: EmbodiedBench (Yang et al., 2025)'s ALFRED (Shridhar et al., 2020) & Habitat (Szot et al., 2021), and VisualAgentBench (Liu et al., 2025)'s OmniGibson (Li et al., 2022) & Minecraft (Microsoft, 2025). (b) Given task, observation, and action history, the previous agents often fail by attempting to pick up the cloth without opening `Cabinet`. On the other hand, INFUSER recognizes that the cloth is likely inside the closed cabinet (failure analysis), generates a corrective plan to open the cabinet first, then successfully retrieves the cloth. Yellow and green boxes represent produced reasoning and observation. Blue boxes indicate successful actions, red indicates failed actions.

**Contribution.** To address this limitation, we propose **INFUSER**: **IN**jecting synthetic **F**ail**U**re for **S**elf-correcting **E**mbodied agents. Our key idea is to enhance agent robustness by explicitly teaching it how to recover from mistakes. We achieve this by augmenting expert demonstrations with synthetically generated failure-and-recovery trajectories. Specifically, we start from ground-truth expert trajectories and inject suboptimal actions (e.g., choosing an incorrect navigation direction) at various decision points. This process creates a diverse set of controlled failure scenarios and their corresponding recovery sequences with minimal human effort. By fine-tuning the VLM on this augmented dataset, which combines both successful and failure-recovery trajectories, INFUSER learns to perceive failures as recoverable states rather than terminal conditions.

We validate the effectiveness of INFUSER through comprehensive evaluations on the EB-ALFRED and EB-Habitat tasks from EmbodiedBench (Yang et al., 2025) and VAB-OmniGibson and VAB-Minecraft from VisualAgentBench (Liu et al., 2025). Our results demonstrate that INFUSER significantly outperforms baseline models trained exclusively on successful trajectories. Notably, INFUSER boosts the performance of the Qwen2.5-VL-7B-Instruct model (Bai et al., 2025) by a substantial margin, elevating the success rate from 18.3% to 47.0% on EB-ALFRED and from 59.7% to 66.3% on EB-Habitat. Beyond task success rates, our error persistence analysis reveals that INFUSER achieves the highest recovery rate (86.1%) among all tested models, including proprietary models, and contains error cascades 67% more effectively than the baseline, demonstrating that failure-aware training fundamentally transforms agent behavior from error avoidance to active recovery. Furthermore, our experiments show that the proposed augmentation strategy is highly scalable, with performance improving monotonically as the ratio of failure-recovery trajectories increases.

## 2 METHOD

In this section, we introduce our synthetic data generation pipeline for training INFUSER, our failure-tolerant VLM (see Figure 1). Our key idea is to generate synthetic failure-recovery dataset, which can be done without human costs (Section 2.1). Then, INFUSER is built by fine-tuning a base VLM on both the original expert and the synthesized failure-recovery trajectories (Section 2.2).

**Problem setup.** We formulate embodied task planning as a sequential decision-making process, i.e., given a natural language instruction $g$, an agent interacts with an environment to achieve the goal. At each timestep $t$, the agent receives a visual observation $o_t \in \mathcal{O}$ and selects an action $a_t \in \mathcal{A}$. The agent's policy $\pi_\theta$ is conditioned on the goal instruction $g$ and the interaction history $h_t = (o_0, a_0, f_0, \ldots, a_{t-1}, f_{t-1}, o_t)$, where $f_i$ is the environmental feedback from action $a_i$. The policy then generates the next action $a_t \sim \pi_\theta(\cdot|h_t, g)$. A complete trajectory $\tau = (o_0, a_0, f_0, \ldots, o_T)$ is a successful trajectory if its final state satisfies the goal conditions.

A key challenge is that the suboptimal actions can lead to failure states that deviate from successful execution paths. In such cases, the agent must recover and replan to accomplish the original objective. Therefore, our goal is to learn a robust policy $\pi_\theta$ that maximizes the task success rate by effectively handling such failures. Therefore, the objective is to:

$$\max_\theta \mathbb{E}_{\tau \sim \pi_\theta}[\mathbf{1}(\text{goal achieved in } \tau)]. \tag{1}$$

This objective requires the policy not only to follow optimal trajectories but also to develop robust recovery capabilities from off-distribution states encountered after failures.

### 2.1 SYNTHETIC FAILURE-RECOVERY DATASET GENERATION

To enhance failure tolerance, our framework generates a synthetic dataset that augments expert demonstrations with failure-recovery trajectories. This process enables the policy model to learn from both successful and corrective action sequences, thereby improving task success rates (see Equation 1). The generation process consists of three main stages.

**Expert Trajectory Processing.** Our method begins by processing expert demonstrations. Specifically, for each ground-truth trajectory $\tau^* = (g, o_0, a_0^*, f_0, \ldots, o_T, a_T^*, f_T)$, where $g$ is the goal and $a_t^*$ is the expert action at timestep $t$, we first generate step-by-step reasoning annotations. We use a large language model (LLM) to generate a natural language explanation $r_t$ for each step. Here, the LLM is prompted with the task goal $g$, the current visual observation $o_t$, the expert action $a_t^*$, and the set of available actions $\mathcal{A}$. The resulting reasoning $r_t$ explains the rationale behind the expert action, its contribution to the overall goal, and its expected outcome. This yields an *augmented* expert trajectory:

$$\tilde{\tau}^* = (g, (o_0, a_0^*, r_0), (o_1, a_1^*, r_1), \ldots, (o_T, a_T^*, r_T)). \tag{2}$$

**Failure Injection.** Next, we inject potential failures into the ground-truth trajectory. To achieve this, we randomly sample timesteps $t_{\texttt{fail}}$ from each expert trajectory $\tilde{\tau}$ with a uniform probability $\rho$. At each sampled step, we prompt an LLM to generate a contextually relevant but suboptimal action $a_{\texttt{fail}}$, conditioned on the history up to that point. The LLM is instructed to generate an action that deviates from $a_t^*$ while remaining within the valid action space $\mathcal{A}$, simulating a realistic mistake:

$$a_{t_{\texttt{fail}}} = \text{LLM}(g, h_{t_{\texttt{fail}}}, a_{t_{\texttt{fail}}}^*, \mathcal{A}) \quad \text{s.t.} \quad a_{t_{\texttt{fail}}} \neq a_{t_{\texttt{fail}}}^* \text{ and } a_{t_{\texttt{fail}}} \in \mathcal{A}. \tag{3}$$

Here, $h_{t_{\texttt{fail}}} = (o_0, a_0^*, \ldots, o_{t_{\texttt{fail}}})$ represents the history leading to the failure point. Injecting failure action within expert trajectory is illustrated in Figure 2

**Recovery Reasoning Generation.** Following a failure injection, we synthesize a corresponding recovery strategy. Using the context of the failure, i.e., including the history $h_{t_{\texttt{fail}}}$, the incorrect action $a_{t_{\texttt{fail}}}$, the resulting observation $o_{t_{\texttt{fail}}}$, and the subsequent sequence of correct actions $\{a_t^*, \ldots, a_T^*\}$, we prompt an LLM to generate a recovery plan:

$$r_{\texttt{recovery}} = \text{LLM}(g, h_{t_{\texttt{fail}}}, a_{t_{\texttt{fail}}}, o_{t_{\texttt{fail}}}, \{a_t^*, \ldots, a_T^*\}). \tag{4}$$

This $r_{\texttt{recovery}}$ analyzes why the incorrect action would prevent goal completion, identifies the necessary corrective actions, and outlines the plan to return to a successful trajectory.

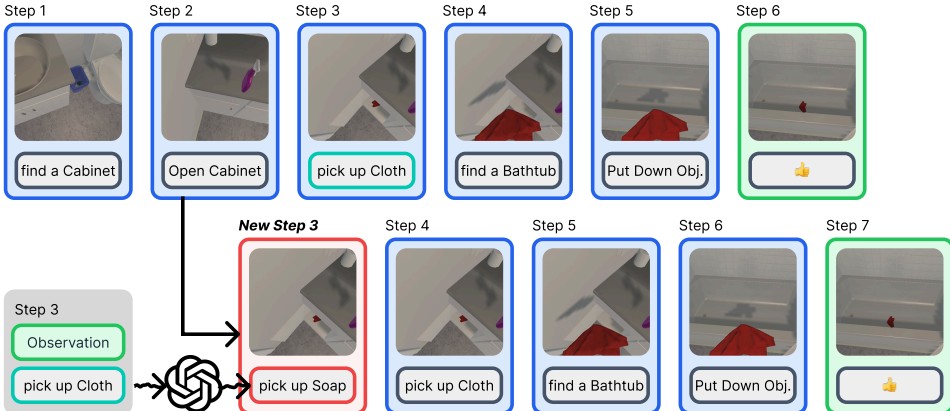

Figure 2: **Illustration of synthetic failure trajectory generation.** Starting from an expert trajectory for "Move cloth to bathtub," we inject a plausible failure at Step 3—the agent mistakenly attempts to pick up soap instead of cloth, simulating object confusion. The subsequent expert actions (pick up cloth, find bathtub, put down) now serve as recovery steps.

## 2.2 TRAINING DATASET CONSTRUCTION

An insight of our method is that to learn effective failure-recovery, the model should be guided by expert actions rather than attempting to learn from potentially suboptimal recovery strategies. Therefore, during training, we provide the ground-truth action $a_t^*$ as an input and train the model to generate the corresponding corrective reasoning that justifies this action. To this end, we construct a unified training dataset $\mathcal{D}_{\texttt{train}}$ that combines successful trajectories with failure-recovery examples.

**Successful Trajectory.** For a standard successful demonstration, each training instance is formulated as predicting the subsequent reasoning and action plan given the history:

$$x_{\texttt{success}}^{(t)} = (g, h_t), \tag{5}$$

$$y_{\texttt{success}}^{(t)} = (r_t, a_{[t:T]}^*). \tag{6}$$

**Failure-Recovery Trajectory.** For failure scenarios, the input is augmented with the expert's recovery action, and the model is trained to generate the rationale behind it. When a failure occurs at timestep $t_{\texttt{fail}}$, the training example is structured as:

$$x_{\texttt{failure}}^{(t_{\texttt{fail}})} = (g, h_{t_{\texttt{fail}}}, a_{t_{\texttt{fail}}}, o_{t_{\texttt{fail}}}, a_{t_{\texttt{fail}}}^*), \tag{7}$$

$$y_{\texttt{failure}}^{(t_{\texttt{fail}})} = (r_{\texttt{recovery}}, a_{[t_{\texttt{fail}}:T]}^*). \tag{8}$$

Here, we provide the expert recovery action $a_{t_{\texttt{fail}}}^*$ directly in the input. With this training scheme, our model focuses on generating corrective reasoning $r_{\texttt{recovery}}$ that explains why $a_{t_{\texttt{fail}}}^*$ is the optimal choice, along with the rest of the future action plan $a_{[t_{\texttt{fail}}:T]}^*$. This form of teacher-forcing on the action space ensures that the model focuses on learning the principles of recovery, rather than being penalized for predicting a suboptimal action while correcting failures. By conditioning on the correct action, we guide the model to develop a robust understanding of failure-recovery logic that can generalize to unseen scenarios at test time. Our final training dataset is the union of both successful and failure-recovery examples:

$$\mathcal{D}_{\texttt{train}} = \{(x_{\texttt{success}}^{(t)}, y_{\texttt{success}}^{(t)})\} \cup \{(x_{\texttt{failure}}^{(t_{\texttt{fail}})}, y_{\texttt{failure}}^{(t_{\texttt{fail}})})\} \tag{9}$$

**Vision-Language Model Fine-tuning.** We fine-tune pre-trained vision-language models (VLMs) on $\mathcal{D}_{\texttt{train}}$ using a standard auto-regressive language modeling objective. Specifically, we initialize our models from Qwen2.5-VL (Bai et al., 2025), aligning with recent approaches in training VLMs for embodied tasks (Luo et al., 2025; Ji et al., 2025; Azzolini et al., 2025). For all experiments, we used a learning rate of $1 \times 10^{-5}$, a batch size of $64$, and trained for $5$ epochs. A complete description of the training configuration is available in the Appendix B.

Table 1: **Performance comparison on EmbodiedBench (EB-ALFRED).** We report the success rates (%) across six task categories and their average. Results with † are reported from Yang et al. (2025). Results with †† are reported from Wu et al. (2025). Qwen2.5-VL-7B* denotes that the model is additionally trained with success trajectories without our failure-recovery trajectories.

| Model | Base | Common | Complex | Visual | Spatial | Long | Average |
|---|---|---|---|---|---|---|---|
| *Proprietary Models* | | | | | | | |
| GPT-4o mini | 34 | 24 | 28 | 26 | 20 | 2 | 22.3 |
| GPT-4o | 64 | 48 | 66 | 46 | 50 | 60 | 55.7 |
| Gemini-1.5-flash † | 44 | 40 | 56 | 42 | 26 | 28 | 39.3 |
| Gemini-2.0-flash | 58 | 46 | 48 | 42 | 40 | 62 | 48.2 |
| Gemini-1.5-Pro † | 70 | 64 | 72 | 58 | 52 | 58 | 62.3 |
| Claude-3.5-Sonnet † | 72 | 66 | 76 | 60 | 58 | 52 | 64.0 |
| Claude-3.7-Sonnet | 70 | 70 | 68 | 66 | 60 | 66 | 66.7 |
| *Open-Source Models (> 7B)* | | | | | | | |
| InternVL3.5-8B | 22 | 16 | 22 | 14 | 10 | 0 | 14.0 |
| InternVL3.5-14B | 38 | 22 | 42 | 26 | 14 | 24 | 27.7 |
| InternVL3.5-38B | 36 | 26 | 38 | 34 | 34 | 30 | 33.0 |
| Qwen2.5-VL-32B | 32 | 28 | 38 | 30 | 38 | 34 | 33.3 |
| Qwen2.5-VL-72B | 42 | 42 | 50 | 42 | 52 | 42 | 45.0 |
| Gemma3-12B | 38 | 32 | 40 | 32 | 18 | 50 | 35.0 |
| *Open-Source Models (= 7B)* | | | | | | | |
| Qwen2-VL-7B | 6 | 0 | 4 | 0 | 0 | 8 | 3.0 |
| Qwen2.5-VL-7B | 8 | 0 | 6 | 4 | 0 | 2 | 3.3 |
| VeBrain-7B | 10 | 4 | 8 | 2 | 0 | 6 | 5.0 |
| RoboBrain-7B | 12 | 6 | 10 | 0 | 0 | 4 | 5.3 |
| Qwen2.5-VL-7B* | 22 | 24 | 20 | 24 | 16 | 4 | 18.3 |
| Reinforce-7B†† | 54 | 42 | 46 | 28 | 38 | 6 | 35.6 |
| **INFUSER-7B (Ours)** | **68** | **50** | **66** | **42** | **40** | **16** | **47.0** |

## 3 EXPERIMENTS

### 3.1 EXPERIMENTAL SETUP

**Datasets.** We evaluate our method, INFUSER, on two benchmarks for embodied multimodal agents that require continuous, multi-step decision making: EmbodiedBench (Yang et al., 2025) and VisualAgentBench (Liu et al., 2025). EmbodiedBench provides environments for household planning tasks across six. We use expert demonstration datasets from ALFRED (Shridhar et al., 2020) for household manipulation and EB-Habitat (Yang et al., 2025) for navigation tasks, both supporting manipulation primitives, navigation commands, and object interactions. VisualAgentBench (Liu et al., 2025) encompasses a variety of visual agent tasks, from which we focus on the Embodied suite: VAB-OmniGibson for manipulation tasks (e.g., "grasp," "put inside") (Li et al., 2022), and VAB-Minecraft, presenting open-world survival and crafting with game-specific actions (e.g., "craft," "mine") (Microsoft, 2025). To construct our primary training data, we augment all expert trajectories from these datasets by synthetically injecting failure steps and generating corresponding failure-recovery data. We refer to Appendix D for details.

**Implementation Details.** We build INFUSER on a state-of-the-art VLM: Qwen2.5-VL (Bai et al., 2025), training a separate instance for each benchmark. All models are fine-tuned using a standard language modeling loss with a learning rate of $1 \times 10^{-5}$ and a batch size of 64. We train for 5 epochs on EmbodiedBench (Yang et al., 2025) and 3 epochs on VisualAgentBench (Liu et al., 2025). For our failure injection strategy, we control the proportion of expert steps to be injected via an augmentation ratio $\rho \in [0, 1.0]$. All failure scenarios and recovery reasoning are generated using GPT-4o (OpenAI, 2024a). We provide detailed hyperparameters in Appendix B.

Table 2: **Performance comparison on EmbodiedBench (EB-Habitat).** We report the success rates (%) across six task categories and their average. Results with † are reported from Yang et al. (2025). Qwen2.5-VL-7B* denotes that the model is additionally trained with success trajectories without our failure-recovery trajectories. The highest scores are shown in **bold**.

| Model | Base | Common | Complex | Visual | Spatial | Long | Average |
|---|---|---|---|---|---|---|---|
| *Proprietary Models* | | | | | | | |
| GPT-4o-mini | 68 | 38 | 28 | 28 | 22 | 10 | 32.3 |
| Gemini-2.0-flash † | 82 | 38 | 38 | 36 | 34 | 26 | 42.3 |
| *Open-Source Models (> 7B)* | | | | | | | |
| InternVL2.5-8B | 48 | 6 | 16 | 10 | 18 | 4 | 17.0 |
| InternVL3.5-8B | 50 | 6 | 24 | 22 | 18 | 8 | 21.3 |
| Qwen2.5-VL-72B | 74 | 28 | 42 | 40 | 24 | 18 | 37.0 |
| *Open-Source Models (= 7B)* | | | | | | | |
| Qwen2.5-VL-7B | 38 | 4 | 12 | 4 | 12 | 6 | 12.7 |
| RoboBrain-7B | 38 | 6 | 18 | 8 | 18 | 4 | 15.3 |
| VeBrain-7B | 44 | 8 | 18 | 18 | 10 | 2 | 16.7 |
| Qwen2.5-VL-7B* | 86 | 50 | **74** | 64 | 42 | 42 | 59.7 |
| **INFUSER-7B (Ours)** | **92** | **60** | 72 | **66** | **48** | **60** | **66.3** |

**Baselines.** We compare INFUSER against proprietary and open-source models. Proprietary models include GPT-4o, GPT-4o-mini (OpenAI, 2024a), Claude-3.7-Sonnet, Claude-3.5-Sonnet (Anthropic, 2025a), Gemini-2.0-flash (Google, 2024), Gemini-1.5-Pro, and Gemini-1.5-flash (Team et al., 2024). Open-source models include Qwen2.5-VL-{7, 32, 72}B (Bai et al., 2025), Qwen2-VL-7B (Wang et al., 2024b), InternVL-3.5-{8, 14, 38}B (Zhu et al., 2025), and Gemma-3-12B (Team et al., 2024).

## 3.2 MAIN RESULTS

**Comparison on EmbodiedBench.** As detailed in Tables 1 and 2, INFUSER achieves new state-of-the-art performance among open-source models on the EmbodiedBench suite. On EB-ALFRED, our approach elevates the success rate of the Qwen2.5-VL-7B model from 3.3% to 47.0%. This result surpasses strong baselines like Reinforce (35.6% success rate) by a significant margin of 11.4 percentage points. In particular, the INFUSER-7B model (47.0%) also outperforms the much larger Qwen2.5-VL-72B baseline (45.0%), demonstrating that targeted failure-recovery training can be more effective than simply scaling the model size. We observe a similar substantial improvement on EB-Habitat, where the success rate surges from 12.7% to 66.3%. Furthermore, across both datasets, our 7B-scale model often outperforms larger, proprietary models such as GPT-4o-mini and Gemini-2.0-flash.

**Comparison on VisualAgentBench.** To validate the broader applicability of our failure-recovery methodology, we extend our evaluation to VisualAgentBench benchmark, utilizing separately trained models (see Table 3). In the VAB-OmniGibson environment, INFUSER elevates the success rate from a 5.5% baseline to 8.8%, marking a substantial 60% relative improvement. A similar trend is observed in VAB-Minecraft, where the success rate increases from 28.4% to 31.4%. The consistent performance enhancements across these distinct benchmarks highlight the robustness of synthetic failure-recovery augmentation, verifying its efficacy for a wide range of embodied agents.

## 3.3 ABLATION STUDY

**Effectiveness of Failure-Recovery Trajectories.** To validate that our performance gains stem from the unique learning signals in failure-recovery data, rather than merely the volume of training samples, we conduct a controlled ablative analysis (see Table 4). We first establish a baseline trained exclusively on 6K successful trajectories, which achieves an 18.3% success rate. We then train our proposed model on a dataset of identical size, composed of 5K successful trajectories and 1K failure-recovery examples. Despite the equivalent data quantity, our failure-augmented model achieves a 33.0% success rate. This constitutes an 80% relative improvement over the success-only baseline,

Table 3: **Performance comparison on VisualAgentBench.** We report success rates (%). Qwen2.5-VL-7B$^*$ denotes that the model is additionally trained with success trajectories without our failure-recovery trajectories.

| Model | OmniGibson | Minecraft |
|---|---|---|
| Qwen2.5-VL-7B | 2.2 | 6.9 |
| Qwen2.5-VL-7B$^*$ | 5.5 | 28.4 |
| **INFUSER-7B** | **8.8** | **31.4** |

Table 4: **Ablation study.** We compare Qwen2.5-VL-7B trained with success data only (Qwen2.5-VL-7B$^*$) and success data with failure-recovery data (INFUSER-7B) in same scale on Embodied-Bench (EB-ALFRED).

| Model | Success | Failure | Avg. |
|---|---|---|---|
| Qwen2.5-VL-7B | - | - | 3.3 |
| Qwen2.5-VL-7B$^*$ | 6K | - | 18.3 |
| **INFUSER-7B** | 5K | 1K | **33.0** |

Figure 3: Ablation study on augmentation ratio $\rho$ on EB-ALFRED.

Table 5: Ablation study on using expert guidance $a^*_{t_{\text{fail}}}$ when constructing failure-recovery trajectories on EmbodiedBench (EB-ALFRED). We report average succes rate (SR; %). Qwen2.5-VL-7B$^*$ denotes that the Qwen2.5-VL-7B additionally trained with success trajectories without our failure-recovery trajectories. We denote the best score as **bold**.

| Model | Expert guidance ($a^*_{t_{\text{fail}}}$) | SR (%) |
|---|---|---|
| Qwen2.5-VL-7B$^*$ | - | 18.3 |
| INFUSER (Variant) | ✗ | 39.3 |
| **INFUSER (Ours)** | ✓ | **47.0** |

confirming that failure-recovery trajectories provide critical information for learning robust recovery policies, which is a signal unavailable in successful demonstrations alone. This highlights that the quality and diversity of the training data are more impactful than sheer quantity in this context.

**Effect of augmentation ratio for failure-recovery trajectories.** We find that performance improves monotonically with the failure augmentation ratio, $\rho \in [0, 1.0]$, as shown in Figure 3. The success rate increases from a baseline of 18.3% ($\rho = 0$) to 47.0% with full augmentation ($\rho = 1.0$), a 157% relative improvement. Even a small fraction of failure data provides substantial gains, with $\rho = 0.1$ boosting performance to 38.0%. These results confirm that exposing the model to diverse failure scenarios is a crucial and scalable method for improving agent capabilities.

**Effectiveness of Expert Guidance.** We examine the importance of expert supervision $a^*_{t_{\text{fail}}}$ (see Equation 7) in Table 5. By training INFUSER with expert ground-truth actions as an input of training example for recovery reasoning targets, INFUSER achieves 47.0% success rate. In contrast, allowing the model to generate its own recovery actions during training yields only 39.3% success rate. This 8.3% percentage point gap shows that expert guidance prevents the model from learning suboptimal recovery strategies, ensuring that failure scenarios lead to principled corrections.

## 4 DISCUSSION

To understand how synthetic failure-recovery training improves embodied agents, we analyze three key aspects of INFUSER's behavior. Section 4.1 examines error persistence patterns, revealing that INFUSER achieves the highest recovery rate (86.1%) among all tested models, including proprietary models, learning active recovery rather than error avoidance. Section 4.2 demonstrates qualitative effect of recovery reasoning. See Figure 4 for example recovery reasoning.

### 4.1 ERROR PERSISTENCE AND RECOVERY ANALYSIS

We analyze error persistence patterns across 300 EB-ALFRED test episodes to understand how failure-aware training affects the agent. We measure four metrics: error streaks (consecutive failures before success), recovery rate (errors corrected within two steps), terminal error rate (errors in final 20% of episodes), and error cascades (sequences of 2+ consecutive errors indicating control loss).

Our analysis, summarized in Table 6, highlights a critical trade-off between task performance and practical deployability in existing embodied agents. Top-performing models, such as Claude-3.7-

Table 6: **Error persistence analysis across models.** We evaluate error handling and recovery capabilities across 300 EB-ALFRED test episodes. Metrics include error streaks (consecutive failures before recovery), recovery rate (errors corrected within 2 steps; %), terminal error rate (%), and error cascades (2+ consecutive errors indicating loss of control). Qwen2.5-VL-7B* denotes that the model is additionally trained with success trajectories without our failure-recovery trajectories.

| | Claude-3.7-Sonnet | Qwen2.5-VL-72B | Qwen2.5-VL-7B* | **INFUSER-7B (Ours)** |
|---|---|---|---|---|
| ***Overall Error Metrics*** | | | | |
| Total Errors (↓) | **764** | 1,126 | 2,469 | 1,733 |
| Total Streaks (↓) | **511** | 904 | 2,136 | 1,484 |
| Avg. Streak Length (↓) | 1.50 | 1.25 | 1.18 | **1.17** |
| ***Recovery Capabilities*** | | | | |
| Recovery Rate (↑) | 65.4 | 73.7 | 77.4 | **86.1** |
| Terminal Error Rate (↓) | 24.3 | 25.0 | 20.2 | **19.8** |
| ***Cascade Analysis*** | | | | |
| Total Cascades (↓) | 83 | **82** | 50 | 164 |
| Avg. Cascade Length (↓) | 4.05 | 3.71 | 7.66 | **2.52** |

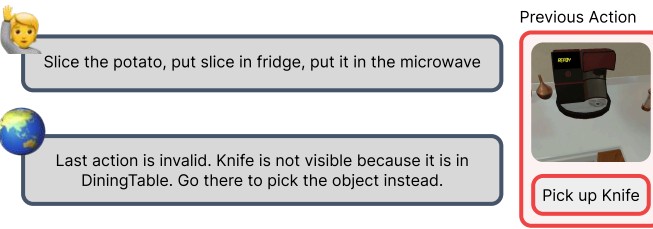

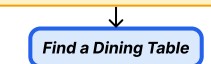

*Success-only model Reasoning*

... begins by locating a countertop, which is a logical starting point for food preparation. **Picking up a knife follows, as it is essential for slicing.** Finding a potato is necessary to proceed with the slicing. Once sliced, the potato needs to be stored in the fridge, ... retrieving the potato slice from the fridge and placing it in the microwave, completing the task. **Each step logically follows the previous one, ensuring the task is completed efficiently.**

→ Pick up Cloth

**INFUSED (Ours) Reasoning**

**The failure occurred because the robot attempted to pick up a knife that was not visible in the current area.** The knife is ... **possibly on a dining table as suggested by the feedback.** ... needs to first locate the knife before proceeding with slicing the potato. The task requires ... finding the knife, ... and then the microwave. **The absence of the knife halted the process, necessitating a search in other locations.**

→ Find a Dining Table

Figure 4: **Comparison of recovery reasoning between a success-only model and INFUSER.** Success-only models provide generic task descriptions that ignore failure and feedback, while INFUSER reasons about the failure cause and suggests corrective actions.

Sonnet, achieve a high task success rate (66.7%) but exhibit poor resilience, with the lowest recovery rate (65.4%) and the longest average error persistence (1.50 steps). As a proprietary API-only model, Claude-3.7-Sonnet cannot be fine-tuned for specific platforms, while Qwen2.5-72B's 72B parameters are computationally prohibitive for edge robotics. Current state-of-the-art models remain unsuitable for deployment scenarios requiring adaptation and failure recovery.

The controlled comparison between our baseline and INFUSER demonstrates that failure-aware training addresses both deployment constraints and resilience requirements within a practical 7B parameter budget. Both models share identical architectures suitable for edge deployment, differing only in training data composition. The inclusion of synthetic failure-recovery trajectories transforms the model's error handling: recovery rate improves from 77.4% to 86.1%, surpassing all tested models. This improvement occurs alongside a 30% reduction in total errors, indicating that failure training develops both preventive and corrective capabilities absent in conventional approaches.

A deeper cascade analysis reveals the mechanism behind INFUSER's robustness. While it encounters more error cascades than the baseline (164 vs. 50), it contains them far more effectively. The

average cascade length for INFUSER is only 2.52 steps, in sharp contrast to the baseline's 7.66 steps. This pattern, combined with INFUSER achieving the lowest terminal error rate (19.8%) across all tested models, shows that its resilience stems from active, learned recovery rather than simple error avoidance. These results show that for embodied agents to be practical and deployable, systematic training on failure scenarios is not just beneficial but essential for achieving the robustness.

## 4.2 LEARNED RECOVERY REASONING

Figure 4 shows that failure-aware training fundamentally changes how the agent reasons. When faced with an error, the success-only model provides a generic plan, ignoring the environmental feedback and the specific failure. In contrast, INFUSER directly addresses the error by identifying its cause ("the knife was not visible"), using feedback to infer a solution, and generating a corrective recovery action ("Find a Dining Table"). This demonstrates that the INFUSER learns to handle failures to be solved through explicit reasoning, a capability absent in models trained solely on ideal trajectories.

## 5 RELATED WORK

**VLMs for Embodied Agents.** Early works adapted pretrained VLMs like CLIP for embodied tasks by grounding natural language goals into visual representations (Dorbala et al., 2022; Gadre et al., 2023; Khandelwal et al., 2022). Subsequent research scaled to large VLMs for high-level planning: ViLaIn (Shirai et al., 2024), GPT-4V for Robotics (Wake et al., 2024), and Voyager (Wang et al., 2024a) leverage LLMs to produce symbolic task plans from demonstrations, while PaLM-E (Driess et al., 2023) provides an end-to-end approach unifying vision and language across diverse embodied tasks. However, most existing approaches focus primarily on leveraging success demonstrations, leaving open the question of how embodied VLM agents can remain robust when confronted with inevitable execution errors.

**Training Embodied Agents.** Training embodied agents typically employs imitation learning through behavior cloning or variants like DAgger (Ross et al., 2011), reinforcement learning with exploration and preference optimization (Tang et al., 2025; Chevalier-Boisvert et al., 2019), or supervised fine-tuning on reasoning-heavy planning tasks (Wu et al., 2025; Shi et al., 2025). Recent work has augmented agents with self-reflection through verbal feedback (Shinn et al., 2023) and structured memory for long-horizon dependencies (Fang et al., 2019). Despite these advances, most paradigms improve success-only policies with limited attention to modeling recovery behaviors after failures.

**Failure-Aware Learning.** Recent research explicitly treats failures as learning signals rather than terminal states. ReplanVLM (Mei et al., 2024) and CMFR (Farag et al., 2025) detect incorrect actions and trigger replanning through error correction modules and multi-stage frameworks with subgoal analysis. ReWiND (Zhang et al., 2025) synthesizes failures by perturbing successful demonstrations for training language-conditioned reward models. While these approaches show that failure data improves robustness, existing methods primarily leverage failures for reward shaping rather than direct training. In contrast, INFUSER systematically injects realistic failures into expert demonstrations with explicit recovery trajectories, directly teaching embodied agents to recover from failure states.

## 6 CONCLUSION

We introduced INFUSER, a framework that transforms brittle embodied agents into resilient systems by augmenting expert demonstrations with synthetically generated failure-recovery trajectories. Our key insight—that agents must learn to treat failures as recoverable states rather than terminal conditions—addresses a fundamental limitation in current training paradigms. INFUSER achieves substantial improvements across diverse benchmarks: $18.3\% \rightarrow 47.0\%$ on EB-ALFRED and $59.7\% \rightarrow 66.3\%$ on EB-Habitat, while attaining the highest recovery rate (86.1%) among all tested models. Remarkably, our 7B parameter model surpasses 72B models, demonstrating that strategic training data composition outweighs model scaling. The synthetic generation approach eliminates costly human demonstrations while enabling practical deployment on resource-constrained platforms. As embodied AI transitions from laboratories to real-world applications, INFUSER's failure-aware training paradigm represents a critical step toward agents that actively recover from errors.

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

## A LLM USAGE DISCLOSURE

This work was developed with assistance from Large Language Models (Claude (Anthropic, 2025b) and GPT-4o (OpenAI, 2024a)) for proofreading, editing, language refinement, and synthetic data generation as described in our methodology. While LLMs supported various aspects of writing and research, all substantive intellectual contributions, experimental results, and scientific claims originate from the human authors, who retain full responsibility for the accuracy and validity of all content. All AI-generated material underwent human review and verification.

## B TRAINING CONFIGURATION AND IMPLEMENTATION DETAILS

### B.1 HYPERPARAMETERS

**Training Configuration.** We implemented INFUSER using LLaMA-Factory (Zheng et al., 2024) as our training framework, leveraging DeepSpeed ZeRO Stage 3 (Rasley et al., 2020) for distributed training optimization along with automatic mixed precision using BF16 and parameter offloading. All models were initialized from Qwen2.5-VL-7B-Instruct Bai et al. (2025) and fine-tuned with the following configuration:

Table 7: Training hyperparameters for INFUSER

| Hyperparameter | Value |
|---|---|
| *Optimization* | |
| Optimizer | AdamW ($\beta_1$=0.9, $\beta_2$=0.999, $\epsilon$=1e-8) |
| Learning rate | $1 \times 10^{-5}$ |
| Learning rate scheduler | Cosine annealing |
| Warmup ratio | 0.1 |
| Weight decay | 0.01 |
| *Training Settings* | |
| Effective batch size | 64 |
| Training epochs (EmbodiedBench (Yang et al., 2025)) | 5 |
| Training epochs (VisualAgentBench (Liu et al., 2025)) | 3 |
| Random seed | 42 |
| Precision | BF16 |
| *Model Configuration* | |
| Base model | Qwen2.5-VL-7B-Instruct |
| Vision tower | Frozen |
| Multi-modal projector | Frozen |
| Language model | Trainable |
| Maximum sequence length | 8,192 tokens |
| Image resolution | 512×512 (max 262,144 pixels) |
| Video resolution | 128×128 (max 16,384 pixels) |

**Framework Details.** Training was conducted using LLaMA-Factory (Zheng et al., 2024), which provides efficient implementations for vision-language model fine-tuning. We utilized the `qwen2_vl` template for consistent formatting of multi-modal inputs and outputs and follow original preprocessing.

## C FAILURE TAXONOMY AND ANALYSIS

We provide comprehensive definitions for each failure type identified in our taxonomy.

### C.1 PLANNING ERRORS

Planning errors occur when agents generate structurally invalid action sequences despite accurate environmental perception.

**Missing Prerequisites**: Attempting actions without satisfying necessary preconditions. The agent executes valid actions but fails to complete enabling steps first, such as opening containers before accessing contents or navigating to objects before manipulation.

**Wrong Action Sequence**: Executing actions in orders that violate task logic or physical constraints. Actions are individually valid but their temporal ordering prevents task completion, such as cleaning objects while contained or placing items before retrieval.

**Spatial Navigation Error**: Failing to establish proper spatial configuration for interaction. The agent attempts manipulation from incorrect positions, fails to account for reachability constraints, or navigates to wrong locations.

### C.2 REASONING ERRORS

Reasoning errors arise from incorrect interpretation of visual input or failure to maintain accurate environmental understanding.

**Perception Error**: Misidentifying task-relevant objects due to visual or semantic confusion. The agent selects incorrect objects that share visual features or semantic categories with intended targets.

**State Tracking Error**: Losing coherence between internal world model and actual environment state. The agent maintains outdated beliefs about object locations or states, failing to update representations based on environmental changes or feedback.

**Container Mismatch**: Selecting incorrect containers or receptacles for object placement. The agent confuses functionally different containers or fails to distinguish between valid and invalid receptacles for specific objects.

**Manipulation Error**: Choosing inappropriate manipulation primitives for object interactions. The agent applies incorrect grasping strategies, force parameters, or manipulation sequences for specific object types.

**Termination Error**: Incorrectly assessing task completion status. The agent declares success with incomplete objectives or continues execution after goal achievement.

These failure modes exhibit hierarchical relationships—planning errors typically manifest as execution failures while reasoning errors produce semantic inconsistencies. Failures often cascade, with initial errors corrupting subsequent decision-making and creating increasingly divergent trajectories from optimal behavior.

### C.3 FAILURE MODE ANALYSIS

Our failure analysis in across 300 EB-ALFRED test episodes reveals that INFUSE addresses systematic planning deficiencies rather than random execution errors. The baseline model exhibits 2,469 failed actions with planning errors dominating (1,532 instances, 62% of failures), primarily driven by missing prerequisite steps where agents attempt object interactions without first opening containers or navigating to proper locations. INFUSE reduces total failures to 1,733 while demonstrating differential recovery effectiveness: planning errors decrease by 41% (1,532 → 905) and object confusion drops by 65% (381 → 132), while reasoning errors show modest 12% improvement (937 → 828). This pattern confirms that systematic errors amenable to learned recovery sequences can be effectively addressed through synthetic failure training.

The error distribution changes reveal the nature of INFUSE's learned behaviors. While step-level success rate improves from 54.0% to 68.4%, manipulation errors actually increase from 556 to 696 instances, suggesting that INFUSE learns to attempt more sophisticated interaction strategies that sometimes fail at execution. This increase in manipulation attempts coupled with overall failure reduction indicates that INFUSE develops persistence strategies—continuing to attempt tasks despite individual action failures rather than abandoning them after initial errors.

Table 8: Failure Mode Analysis on EB-ALFRED (300 test episodes)

| Failure Type | Claude-3.7 | Qwen2.5-72B | Success (7B) | INFUSE (7B) |
|---|---|---|---|---|
| Total Steps | 4,587 | 5,029 | 5,367 | 5,490 |
| Total Failures | 764 | 1,126 | 2,469 | 1,733 |
| Task Success Rate (%) | 66.7 | 45.0 | 18.3 | 47.0 |
| Step Success Rate (%) | 83.3 | 77.6 | 54.0 | 68.4 |
| Avg. Failures/Episode | 2.5 | 3.8 | 8.2 | 5.8 |
| *Error Type Distribution* | | | | |
| Planning Errors | 241 | 251 | 1,532 | 905 |
| Reasoning Errors | 523 | 875 | 937 | 828 |
| *Error Subtype Breakdown* | | | | |
| Missing Prerequisites | 231 | 243 | 1,451 | 873 |
| Manipulation Errors | 459 | 777 | 556 | 696 |
| Object Confusion | 64 | 98 | 381 | 132 |
| Navigation Errors | 10 | 8 | 81 | 32 |

## D DATASET DETAILS

### D.1 EMBODIEDBENCH

We construct dataset for expert demonstration from released data from ALFRED (Shridhar et al., 2020), following Wu et al. (2025). From release data, we collect a total of 6,574 expert trajectory with total 62,255 steps. Out of this total steps, we exclude the first and last step to be sampled, with the result total 49,045 failure trajactory, excluding a few malicious data. For EB-Habitat, we use published expert trajectory from Yang et al. (2025). From total 3,235 trajactory, we filter out 1,651 only successful trajactories. Among 1,651 success trajactories, we generated 4904 failure trajactory.

### D.2 VISUALAGENTBENCH

For VisualAgentBench, we use training dataset from VisualAgentBench (Liu et al., 2025). VAB-OmniGibson contains 872 trajactory with total 20,153 steps, and VAB-Minecraft contains 382 train trajactory with total 5,197 train step. From their training data, we generate 19165 and 4745 failure-recovery trajactory, respectively.

## E ADDITIONAL DETAILS FOR EMBODIEDBENCH

### E.1 OVERVIEW OF EMBODIEDBENCH FRAMEWORK

EmbodiedBench establishes a pioneering evaluation paradigm for vision-language models operating within interactive embodied environments. This comprehensive framework transcends conventional static benchmarks by immersing agents in dynamic, physically grounded simulations where multimodal perception, semantic reasoning, and sequential action execution must harmoniously integrate. The benchmark encompasses four distinct simulation environments supporting over 1,100 meticulously designed tasks, ranging from atomic actions to complex multi-step planning sequences that challenge current model capabilities.

The framework's architecture emphasizes ecological validity through realistic task scenarios grounded in everyday activities, while maintaining computational tractability for systematic evaluation. By bridging the gap between controlled laboratory conditions and real-world deployment requirements, EmbodiedBench provides essential infrastructure for advancing embodied artificial intelligence.

### E.2 EB-ALFRED: HOUSEHOLD TASK ENVIRONMENT

**Environment Specifications.** EB-ALFRED, built upon the ALFRED dataset foundation and AI2-THOR simulator, delivers a sophisticated household interaction platform featuring eight core action

primitives: object manipulation (*pick up*, *put down*), spatial navigation (*find*), state modifications (*open/close*, *turn on/off*), and object processing (*slice*, *clean*). The environment provides first-person RGB visual observations complemented by structured textual feedback, enabling agents to develop robust planning strategies through iterative refinement.

**Technical Enhancements.** Distinguished from its predecessor, EB-ALFRED incorporates critical architectural improvements: support for multiple instances of identical object types within scenes, consolidated action sets with unified manipulation commands, and dynamically adjustable action spaces spanning 171 to 298 distinct actions based on scene configuration. These enhancements create a more naturalistic evaluation environment that better approximates real-world embodied reasoning challenges.

**Task Complexity and Distribution.** EB-Alfred environment encompasses 6,535 training trajectories spanning six specialized categories: base tasks establishing fundamental competencies, common sense reasoning scenarios requiring implicit world knowledge, complex instruction processing with verbose or ambiguous directives, spatial awareness challenges demanding 3D relationship understanding, visual appearance recognition requiring fine-grained discrimination, and long horizon planning sequences exceeding 15 coordinated actions. Indeed, we do not use this 6,535 training trajectories. Following Wu et al. (2025), we directly used trajactory released from ALFRED (Shridhar et al., 2020).

### E.3 EB-Habitat: Spatial Reasoning Environment

**Environment Architecture.** EB-Habitat, extending the Language Rearrangement benchmark within the Habitat 2.0 framework, emphasizes spatial reasoning and object manipulation within photorealistic indoor scenes. The environment focuses on five core competencies: goal-directed navigation, object grasping, precise placement, container manipulation, and spatial relationship understanding. Navigation constraints require agents to approach receptacle-type destinations exclusively, necessitating sophisticated scene understanding capabilities.

**Instruction Templates and Diversity.** The environment provides 282 distinct instruction templates that challenge agents' spatial reasoning abilities through relative positioning requirements, containment relationships, and efficient path planning scenarios. This template diversity ensures comprehensive evaluation of spatial cognitive capabilities across varied linguistic formulations and environmental configurations.

### E.4 Task Taxonomy and Evaluation Dimensions

Both EmbodiedBench environments employ a systematic six-category taxonomy designed to isolate and assess specific cognitive capabilities across embodied reasoning scenarios. This standardized categorization enables comprehensive evaluation of agent competencies while facilitating meaningful performance analysis across different environmental contexts.

**Evaluation Categories.**

- **Base Tasks:** Fundamental scenarios establishing core embodied reasoning competencies through standard object interactions and straightforward goal achievement under moderate complexity constraints.
- **Common Sense Reasoning:** Scenarios requiring implicit world knowledge application and ambiguous reference resolution through understanding of typical object uses, default locations, and everyday behavioral patterns.
- **Complex Instruction Processing:** Linguistically challenging directives featuring verbose descriptions, nested clauses, or extraneous information requiring salient goal extraction from naturalistic, unstructured language.
- **Spatial Awareness:** Three-dimensional relationship comprehension including relative positioning, containment hierarchies, and navigational constraints requiring accurate spatial mental model maintenance.

- **Visual Appearance Recognition:** Fine-grained visual discrimination based on object attributes including color, texture, material, and shape, testing perceptual acuity beyond simple classification.

- **Long Horizon Planning:** Extended sequences requiring 15+ coordinated actions evaluating temporal reasoning, state tracking, and plan maintenance over extended interaction periods.

# F    ADDITIONAL DETAILS FOR VISUALAGENTBENCH EMBODIED SUITE

## F.1    OVERVIEW OF VISUALAGENTBENCH EMBODIED FRAMEWORK

VisualAgentBench (VAB) establishes a comprehensive evaluation paradigm for Multimodal Large Language Models (MLLMs) as visual foundation agents across interactive embodied scenarios. This pioneering framework transcends traditional static evaluation methodologies by requiring agents to process egocentric visual observations, interpret environmental feedback, and execute coherent action sequences within visually rich, physically grounded simulations. The embodied suite encompasses two meticulously designed environments that challenge agents' capabilities in spatial reasoning, object manipulation, and sequential decision-making.

The framework's distinctive approach addresses fundamental gaps in existing evaluation methodologies by providing both high-quality test sets and comprehensive training trajectories essential for developing robust visual agents. By enabling systematic assessment of both proprietary and open-source MLLMs through standardized interfaces, VAB facilitates rigorous comparison and advancement of embodied multimodal intelligence.

## F.2    VAB-OMNIGIBSON: HOUSEHOLD EMBODIED INTELLIGENCE

**Environment Specifications.**    VAB-OmniGibson, built upon the OmniGibson simulator (Li et al., 2022), delivers photorealistic rendering and sophisticated physics simulation for household environments. The platform features 20 distinct high-level actions optimized for semantic planning evaluation: object manipulation primitives (`grasp`, `put_inside`, `put_on_top`), navigation commands (`move`, `move_to_room`), camera control operations, and state modification actions (`open`, `close`, `toggle_on/off`, `cook`, `heat`, `freeze`).

**Technical Architecture.**    The environment transcends limitations of existing household benchmarks through carefully curated action abstraction that enables MLLMs to focus on high-level planning capabilities rather than low-level motor control. Tasks are grounded in specific scene configurations with randomized object positions, ensuring robust evaluation across varying initial conditions. BEHAVIOR Domain Definition Language (BDDL) specifications define precise goal conditions, enabling deterministic success evaluation while maintaining ecological validity.

**Task Complexity and Distribution.**    VAB-OmniGibson encompasses 181 test instances derived from 45 carefully selected activity prototypes spanning simple object relocations to complex multi-step procedures including meal preparation and household organization. The training infrastructure comprises 872 trajectories collected through hybrid methodology: 785 trajectories from rule-based solvers across 901 training instances, supplemented by 87 MLLMs bootstrapping trajectories from GPT-4V, introducing natural variability in problem-solving approaches.

## F.3    VAB-MINECRAFT: OPEN-WORLD EMBODIED REASONING

**Environment Specifications.**    VAB-Minecraft, extending the MineRL framework (Minerllabs, 2025) with JARVIS-1 integration (Wang et al., 2024c), provides open-world scenarios requiring exploration, resource gathering, and multi-step crafting procedures. The environment presents unique challenges through procedural world generation, diverse biome navigation, and complex item dependency chains spanning six material tiers from wood through netherite. The platform supports both peaceful and hostile mob interactions, necessitating adaptive planning strategies.

**Action Space and Capabilities.**    The environment implements six high-level actions tailored for multi-modal large language models (MLLMs) agents: `craft` and `smelt` for item creation, `equip` for tool management, `teleport_to_spawn` for navigation reset, `look_up` for recipe

information retrieval, and `execute` for invoking specialized low-level controllers. This abstraction enables MLLMs to leverage natural language understanding for complex instruction following while delegating fine-grained control to dedicated models.

**Task Diversity and Scale.** VAB-Minecraft features 116 test tasks encompassing diverse resource types including 11 plant varieties, 4 animal types, and 6 hostile mob categories. Tasks evaluate progressive skill acquisition from basic resource gathering to advanced item synthesis requiring multiple intermediate crafting steps. Training data comprises 382 successful trajectories: 176 from GPT-4-turbo (OpenAI, 2024b) on newly designed tasks and 206 from GPT-4o with JARVIS-1 memory augmentation, capturing both exploratory and optimized solution strategies.

## F.4 EVALUATION METHODOLOGY AND PERFORMANCE METRICS

**Assessment Framework.** Both environments employ task success rate as the primary evaluation metric, measuring completion within prescribed interaction limits (100 rounds per environment). Success determination utilizes environment-specific criteria: BDDL goal satisfaction for OmniGibson and target item acquisition for Minecraft. Interactive evaluation protocols require agents to engage directly with simulated environments, processing visual observations and environmental feedback at each timestep while maintaining temporal consistency across extended sequences.

**Statistical Distribution.** VAB-OmniGibson provides 181 test instances across 45 activity types with 872 training trajectories averaging 20.15 steps per sequence across 20-dimensional action space. VAB-Minecraft offers 116 test instances with 382 training trajectories averaging 13.6 steps per sequence across 6-dimensional action space, covering comprehensive crafting hierarchies and 21 distinct resource types distributed across multiple material progression tiers.

# G TECHNICAL IMPLEMENTATION DETAILS

## G.1 SYNTHETIC FAILURE GENERATION METHODOLOGY

**Uniform Sampling Strategy.** Our synthetic failure injection employs uniform random sampling across expert trajectory steps, ensuring unbiased coverage of potential failure scenarios without predetermined error type distributions. The our framework samples $\rho$ of eligible steps from each trajectory, excluding initial and final steps to preserve task initiation and completion integrity.

**Failure Candidate Generation Pipeline.** Throughout our work, we use GPT-4o (OpenAI, 2024a) (`gpt-4o-2024-08-06`) model for all data annotation purpose. The failure injection process operates through a streamlined pipeline: (1) trajectory preprocessing identifies eligible steps for failure injection, (2) uniform random sampling selects candidate steps based on a sample rate $\rho$, (3) contextual alternative action generation utilizes GPT-4o to suggest plausible but suboptimal actions given environmental context and available action spaces, and (4) validation filtering removes actions that violate environmental constraints or action space definitions (e.g. alternative action is not in action space, returning same action).

## H  EXAMPLE TRAJECTORY IN EB-ALFRED BENCHMARK

| Visual Input | VLM Analysis | Action & Feedback |
|---|---|---|
| **Task:** Rinse off a ladle and move it to the table. | | |
|  | **Visual:** The image shows a section of a kitchen with a large dark-colored cabinet door partially visible on the left side. There is a metal sink located at the
**Reasoning:** The task requires rinsing a ladle and moving it to a table. The action sequence begins by locating a countertop, which i
**Plan:** 1. Locate a countertop where the ladle might be found | **ID:** 39
**Action:** find a CounterTop
**Feedback:** Last action executed successfully.
**Success:** ✓ |
|  | **Visual:** The image shows an outdoor area with a concrete surface, possibly a patio or garden path. There are shadows cast across the surface, likely from nearb
**Reasoning:** The failure occurred because the ladle is located on a different countertop that was not initially considered. The actio
**Plan:** 1. Locate the CounterTop_2 where the ladle is placed | **ID:** 39
**Action:** find a CounterTop
**Feedback:** Last action executed successfully.
**Success:** ✓ |
|  | **Visual:** The image shows a kitchen environment with a countertop next to a stove. On the countertop, there are two metal cups and a wooden cutting board. A pot
**Reasoning:** The failure occurred because the ladle was not visible when the robot attempted to pick it up from the countertop. This
**Plan:** 1. Locate where the ladle is stored by performing a find action to discover its exact position | **ID:** 109
**Action:** pick up the Ladle
**Feedback:** Last action executed successfully.
**Success:** ✓ |
|  | **Visual:** The image shows a kitchen environment with a countertop next to a stove. On the countertop, there are two metal cups and a wooden cutting board. A pot
**Reasoning:** The failure occurred because the ladle was not visible when the robot attempted to pick it up from the countertop. This
**Plan:** 1. Locate where the ladle is stored by performing a find action to discover its exact position | **ID:** 155
**Action:** turn on the Faucet
**Feedback:** Last action executed successfully.
**Success:** ✓ |
|  | **Visual:** The image shows a kitchen environment with a countertop next to a stove. On the countertop, there are two metal cups and a wooden cutting board. A pot
**Reasoning:** The failure occurred because the ladle was not visible when the robot attempted to pick it up from the countertop. This
**Plan:** 1. Locate where the ladle is stored by performing a find action to discover its exact position | **ID:** 109
**Action:** pick up the Ladle
**Feedback:** Last action executed successfully.
**Success:** ✓ |

Figure 5: **Example execution result of INFUSER  in EB-ALREAD**, Sampled total 6 steps out of full trajectory with interval of 3 steps

| Visual Input | VLM Analysis | Action & Feedback |
|---|---|---|
| **Task:** wash the ladle and put it back on the table | | |
| | **Visual:** The image shows a part of a room with a smooth, beige wall and a small section of what appears to be a floor or furniture edge. There are no visible o
**Reasoning:** The task requires washing a ladle and returning it to the table. The action sequence begins by locating the dining table
**Plan:** 1. Locate the dining table to find the ladle | **ID:** 18
**Action:** find a DiningTable
**Feedback:** Last action executed successfully.
**Success:** ✓ |
| | **Visual:** The image shows a part of a room with a smooth, beige wall and a small section of what appears to be a floor or furniture edge. There are no visible o
**Reasoning:** The task requires washing a ladle and returning it to the table. The action sequence begins by locating the dining table
**Plan:** 1. Locate the dining table to find the ladle | **ID:** 109
**Action:** pick up the Ladle
**Feedback:** Last action executed successfully.
**Success:** ✓ |
| | **Visual:** The image shows a part of a room with a smooth, beige wall and a small section of what appears to be a floor or furniture edge. There are no visible o
**Reasoning:** The task requires washing a ladle and returning it to the table. The action sequence begins by locating the dining table
**Plan:** 1. Locate the dining table to find the ladle | **ID:** 133
**Action:** put down the object in hand
**Feedback:** Last action executed successfully.
**Success:** ✓ |
| | **Visual:** The image shows a part of a room with a smooth, beige wall and a small section of what appears to be a floor or furniture edge. There are no visible o
**Reasoning:** The task requires washing a ladle and returning it to the table. The action sequence begins by locating the dining table
**Plan:** 1. Locate the dining table to find the ladle | **ID:** 156
**Action:** turn off the Faucet
**Feedback:** Last action executed successfully.
**Success:** ✓ |
| | **Visual:** The image shows a part of a room with a smooth, beige wall and a small section of what appears to be a floor or furniture edge. There are no visible o
**Reasoning:** The task requires washing a ladle and returning it to the table. The action sequence begins by locating the dining table
**Plan:** 1. Locate the dining table to find the ladle | **ID:** 18
**Action:** find a DiningTable
**Feedback:** Last action executed successfully.
**Success:** ✓ |
| | **Visual:** The image shows a part of a room with a smooth, beige wall and a small section of what appears to be a floor or furniture edge. There are no visible o
**Reasoning:** The task requires washing a ladle and returning it to the table. The action sequence begins by locating the dining table
**Plan:** 1. Locate the dining table to find the ladle | **ID:** 133
**Action:** put down the object in hand
**Feedback:** Last action executed successfully.
**Success:** ✓ |

Figure 6: **Example execution result of INFUSER in EB-ALREAD**, Sampled total 6 steps out of full trajectory with interval of 3 steps

# I    PROMPT USED FOR DATASET GENERATION

---

**Failure Candidate Prompt for EmbodiedBench**

```
<system>
You are an expert at analyzing robot task execution trajectories and identifying plausible
    alternative actions that could be taken at specific points in the sequence.

Your task is to suggest ONE alternative action that is:
1. Plausible but less optimal than the expert action
2. Would potentially lead to a failure or inefficiency
3. Still makes some logical sense given the context
4. Different from the current expert action

Consider common failure scenarios:
- Trying to pick up objects that don't exist or are in closed receptacles
- Attempting actions when robot is already holding something
- Trying to open/close objects that are already in that state
- Looking for objects in wrong locations
- Attempting to put objects in inappropriate receptacles

Return your response as a JSON object with:
{
    "alternative_action": "the alternative action name",
    "reasoning": "brief explanation of why this alternative might be chosen and why it would be less
        optimal"
}

IMPORTANT: The alternative action MUST be from the provided action space list.

<user>
"Task: {task_instruction}

Expert trajectory up to this point:
{previous_trajectory}

Current expert action (step {current_index}/{total_action_count}): {current_action}

Following expert actions:
{remaining_expert_actions}

Based on the task and trajectory context, suggest ONE plausible alternative action that could be
    taken instead of "{current_action}".

Available actions:
{action_list}

The alternative action must be from the above list and should be different from the current expert
    action.
```

---

## Expert Reasoning Prompt for EmbodiedBench

```
<system>
You are an expert at analyzing visual scenes and generating detailed reasoning for robot task
    planning in household environments.

Your task is to generate reasoning that explains WHY a specific sequence of actions is
    appropriate for accomplishing a given task, based on the visual observation and task
    instruction.

You will receive:
1. An image showing the initial state of the environment
2. A task instruction describing what needs to be accomplished
3. An expert action sequence that successfully completes the task

You must generate a JSON response with the following structure:
{
  "visual_state_description": "Detailed description of what you observe in the image, including
      objects, their positions, and spatial relationships",
  "reasoning_and_reflection": "Logical reasoning explaining why this action sequence is
      appropriate for the task, considering the visual state and task requirements",
  "language_plan": "Step-by-step natural language description of the plan, explaining what each
      action accomplishes",
  "executable_plan": [List of actions with action_id and action_name]
}

Important guidelines:
- The visual_state_description should accurately describe what is visible in the image
- The reasoning_and_reflection should explain the logic behind choosing these specific actions
- The language_plan should break down the task into clear, understandable steps
- Each action in executable_plan must have both action_id and action_name
- Be precise and avoid unnecessary details
- Focus on explaining WHY these actions lead to task completion
- Consider object locations, accessibility, and task requirements in your reasoning

<user>
Task Instruction: {task_instruction}

Expert Action Sequence: {expert_action_sequence}

Based on the image and task instruction above, generate detailed reasoning for why this specific
    action sequence effectively accomplishes the task. Your response should explain the logic
    behind each action choice and how they work together to complete the task.

Remember to format your response as a JSON object with the required fields:
    visual_state_description, reasoning_and_reflection, language_plan, and executable_plan

The executable_plan has been mapped to the following action IDs: {
    action_id_mapped_expert_sequence}

Please incorporate this exact executable_plan in your response.
```

1242
1243
1244
1245
1246
1247
1248
1249
1250

**Failure Recovery Prompt for EmbodiedBench**

1251 You are an expert at analyzing visual scenes and generating recovery strategies for robot task
planning in household environments.

1252
1253 Your task is to analyze a failure scenario and generate reasoning that explains:
1. WHY the previous actions failed based on the visual observation and environment feedback
1254 2. HOW to recover and complete the task successfully from the current state

1255 You will receive:
1. An image showing the current state after the failure occurred
1256 2. A task instruction describing what needs to be accomplished
3. An action history showing what was attempted and the environment feedback
1257 4. Information about which action failed and why

1258 Available Actions (0-{action_space_size}):
1259 {action_list}

1260 You must generate a JSON response with the following structure:
{{
1261   "visual_state_description": "Detailed description of what you observe in the current image
after the failure, including objects, their positions, and the robot's current state",
1262   "reasoning_and_reflection": "Analysis of why the failure occurred based on the visual state
and environment feedback, plus logical reasoning for the recovery strategy",
1263   "language_plan": "Step-by-step natural language description of the recovery plan, explaining
what each remaining action will accomplish",
1264   "executable_plan": [List of actions with action_id and action_name to complete the task from
current state]
1265 }}

1266
1267 Important guidelines for failure recovery:
- The visual_state_description should capture the state AFTER the failure occurred
1268 - The reasoning_and_reflection must explain WHY the failure happened and justify the recovery
approach
1269 - Consider environment feedback messages carefully – they indicate specific issues
- The recovery plan should address the root cause of the failure
1270 - Avoid repeating the same failed action sequence
- Be precise about the current robot state (holding object, location, etc.)
1271 - Focus on explaining WHY the recovery actions lead to task completion

1272 Task Instruction: {task_instruction}

1273 Action History with Environment Feedback:
1274 {action_history}

1275 Failed Action Details:
- Action attempted: {failed_action_name} (id: {failed_action_id})
1276 - Failure reason: {env_feedback}

1277 Based on the image showing the current state after this failure, generate detailed reasoning for
:
1278 1. Why this specific failure occurred given the visual state and task requirements
2. How to recover and successfully complete the task from this point
1279

1280 Your recovery plan should complete the remaining task objectives.
The expert demonstration shows these actions would complete the task from this point:
1281 [
{remaining_expert_action_sequence}
1282 ]

1283 Use this as guidance for your recovery plan.

1284 Remember to format your response as a JSON object with the required fields:
visual_state_description, reasoning_and_reflection, language_plan, and executable_plan.
1285

1286 Each action in executable_plan must have both action_id (0-{action_space_size}) and action_name
from the available actions.

1287
1288
1289
1290
1291
1292
1293
1294
1295

1296
1297
1298
1299
1300
1301
1302
1303
1304
1305
1306
1307
1308
1309
1310
1311
1312
1313
1314
1315
1316
1317
1318
1319
1320
1321
1322
1323
1324
1325
1326
1327
1328
1329
1330
1331
1332
1333
1334
1335
1336
1337
1338
1339
1340
1341
1342
1343
1344
1345
1346
1347
1348
1349

**Failure Recovery SFT Prompt for EmbodiedBench**

```
You are an expert at analyzing visual scenes and generating recovery strategies for robot task
    planning in household environments.

Your task is to analyze a failure scenario and generate reasoning that explains:
1. WHY the previous actions failed based on the visual observation and environment feedback
2. HOW to recover and complete the task successfully from the current state

You will receive:
1. An image showing the current state after the failure occurred
2. A task instruction describing what needs to be accomplished
3. An action history showing what was attempted and the environment feedback
4. Information about which action failed and why

You must generate a JSON response with the following structure:
{{
  "visual_state_description": "Detailed description of what you observe in the current image
      after the failure, including objects, their positions, and the robot's current state",
  "reasoning_and_reflection": "Analysis of why the failure occurred based on the visual state
      and environment feedback, plus logical reasoning for the recovery strategy",
  "language_plan": "Step-by-step natural language description of the recovery plan, explaining
      what each remaining action will accomplish",
  "executable_plan": [List of actions with action_id and action_name to complete the task from
      current state]
}}

Important guidelines for failure recovery:
- The visual_state_description should capture the state AFTER the failure occurred
- The reasoning_and_reflection must explain WHY the failure happened and justify the recovery
    approach
- Consider environment feedback messages carefully - they indicate specific issues
- The recovery plan should address the root cause of the failure
- Avoid repeating the same failed action sequence
- Be precise about the current robot state (holding object, location, etc.)
- Focus on explaining WHY the recovery actions lead to task completion

Task Instruction: {task_instruction}

Action History with Environment Feedback:
{action_history}

Failed Action Details:
- Action attempted: {failed_action_name} (id: {action_id})
- Failure reason: {env_feedback}

Based on the image showing the current state after this failure, generate detailed reasoning for
    :
1. Why this specific failure occurred given the visual state and task requirements
2. How to recover and successfully complete the task from this point

Your recovery plan should complete the remaining task objectives.
The expert demonstration shows these actions would complete the task from this point:
[
{remaining_expert_action_sequence}
]

Use this as guidance for your recovery plan.

Remember to format your response as a JSON object with the required fields:
    visual_state_description, reasoning_and_reflection, language_plan, and executable_plan.

Each action in executable_plan must have both action_id (0-{action_space_size}) and action_name
    from the available actions.

Available Actions (0-{action_space_size}):
{action_list}
```

## SFT Prompt for EB-ALFRED

## You are a robot operating in a home. Given a task, you must accomplish the task using a defined set of actions to achieve the desired outcome.

## Action Descriptions and Validity Rules
- Navigation: Parameterized by the name of the receptacle to navigate to. So long as the receptacle is present in the scene, this skill is always valid
- Pick: Parameterized by the name of the object to pick. Only valid if the robot is close to the object, not holding another object, and the object is not inside a closed receptacle.
- Place: Parameterized by the name of the receptacle to place the object on. Only valid if the robot is close to the receptacle and is holding an object.
- Open: Parameterized by the name of the receptacle to open. Only valid if the receptacle is closed and the robot is close to the receptacle.
- Close: Parameterized by the name of the receptacle to close. Only valid if the receptacle is open and the robot is close to the receptacle.

## The available action id (0 ~ {action_space_size}) and action names are:
{action_list}

{in_context_examples}

## Guidelines
1. **Output Plan**: Avoid generating empty plan. Each plan should include no more than 20 actions.
2. **Visibility**: If an object is not currently visible, use the "Navigation" action to locate it or its receptacle before attempting other operations.
3. **Action Validity**: Make sure match the action name and its corresponding action id in the output.
 Avoid performing actions that do not meet the defined validity criteria.
4. **Prevent Repeating Action Sequences**: Do not repeatedly execute the same action or sequence of actions.
 Try to modify the action sequence because previous actions do not lead to success.
5. **Multiple Instances**: There may be multiple instances of the same object, distinguished by an index following their names, e.g., cabinet 2, cabinet 3. You can explore these instances if you do not find the desired object in the current receptacle.
6. **Reflection on History and Feedback**: Use interaction history and feedback from the environment to refine and enhance your current strategies and actions. If the last action is invalid, reflect on the reason, such as not adhering to action rules or missing preliminary actions, and adjust your plan accordingly.

## Now the human instruction is: {task_instruction} You are supposed to output in json. You need to describe current visual state from the image, output your reasoning steps and plan. At the end, output the action id (0 ~ {action_space_size}) from the available actions to execute.

**SFT Prompt for EB-Habitat**

```
## You are a robot operating in a home. Given a task, you must accomplish the task using a
     defined set of actions to achieve the desired outcome.

## Action Descriptions and Validity Rules
- Navigation: Parameterized by the name of the receptacle to navigate to. So long as the
     receptacle is present in the scene, this skill is always valid
- Pick: Parameterized by the name of the object to pick. Only valid if the robot is close to the
     object, not holding another object, and the object is not inside a closed receptacle.
- Place: Parameterized by the name of the receptacle to place the object on. Only valid if the
     robot is close to the receptacle and is holding an object.
- Open: Parameterized by the name of the receptacle to open. Only valid if the receptacle is
     closed and the robot is close to the receptacle.
- Close: Parameterized by the name of the receptacle to close. Only valid if the receptacle is
     open and the robot is close to the receptacle.

## The available action id (0 ~ {action_space_size}) and action names are:
{action_list}

{in_context_examples}

## Guidelines
1. **Output Plan**: Avoid generating empty plan. Each plan should include no more than 20
     actions.
2. **Visibility**: If an object is not currently visible, use the "Navigation" action to locate
     it or its receptacle before attempting other operations.
3. **Action Validity**: Make sure match the action name and its corresponding action id in the
     output.
 Avoid performing actions that do not meet the defined validity criteria.
4. **Prevent Repeating Action Sequences**: Do not repeatedly execute the same action or sequence
     of actions.
 Try to modify the action sequence because previous actions do not lead to success.
5. **Multiple Instances**: There may be multiple instances of the same object, distinguished by
     an index following their names, e.g., cabinet 2, cabinet 3. You can explore these
     instances if you do not find the desired object in the current receptacle.
6. **Reflection on History and Feedback**: Use interaction history and feedback from the
     environment to refine and enhance your current strategies and actions. If the last action
     is invalid, reflect on the reason, such as not adhering to action rules or missing
     preliminary actions, and adjust your plan accordingly.

## Now the human instruction is: {task_instruction} You are supposed to output in json. You need
     to describe current visual state from the image, output your reasoning steps and plan. At
     the end, output the action id (0 ~ {action_space_size}) from the available actions to
     execute.
```

## Failure Candidate Prompt for VAB-OmniGibson

```
<system>
You are an expert at analyzing robot task execution trajectories and identifying plausible
    alternative actions that could be taken at specific points in the sequence.

Your task is to suggest ONE alternative action that is:
1. Plausible but less optimal than the expert action
2. Would potentially lead to a failure or inefficiency
3. Still makes some logical sense given the context
4. Different from the current expert action

Consider common failure scenarios in household robotics:
- Trying to grasp objects that are not within reach
- Attempting actions when robot is already holding something
- Trying to put objects in inappropriate or closed containers
- Attempting to open/close objects that don't support those actions
- Looking for objects in wrong locations
- Attempting to manipulate objects without proper prerequisites

IMPORTANT: Only suggest core functional actions from this list:
grasp, move, put_inside, put_on_top, put_under, put_next_to, cook, burn, freeze, heat, open,
    close, toggle_on, toggle_off, move_to_room

DO NOT suggest camera, turn, or perception actions as these are basic navigation aids.

Return your response as a JSON object with:
{
    "alternative_action": "the alternative action in the same format as provided actions",
    "reasoning": "brief explanation of why this alternative might be chosen and why it would be
        less optimal"
}

IMPORTANT: The alternative action MUST follow the function format (e.g., grasp(obj), move(obj),
    etc.) and use only the core functional actions listed above

<user>
Task Goal: {task_instruction}

Environmental Context:
- Current Room: {current_room}
- At Hand Object: {at_hand_object}
- Action Feedback: {action_feedback}
- Reachable Rooms: {reachable_rooms}

Expert action sequence up to this point:
{previous_actions}

Current expert action (step {current_index}/{total_action_count}): {current_action}

Following expert actions:
{remaining_expert_actions}

Based on the task, environmental context, and trajectory context, suggest ONE plausible
    alternative action that could be taken instead of "{current_action}".

Available core functional action templates:
    'grasp(obj)', 'move(obj)', 'put_inside(obj1, obj2)', 'put_on_top(obj1, obj2)',
    'put_under(obj1, obj2)', 'put_next_to(obj1, obj2)', 'cook(obj)', 'burn(obj)',
    'freeze(obj)', 'heat(obj)', 'open(obj)', 'close(obj)', 'toggle_on(obj)',
    'toggle_off(obj)', 'move_to_room(room)'

The alternative action must follow one of these function templates and should be different from
    the current expert action.
```

1512
1513
1514
1515
1516
1517
1518
1519
1520
1521
1522
1523
1524
1525
1526
1527
1528
1529
1530
1531
1532
1533
1534
1535
1536
1537
1538
1539
1540
1541
1542
1543
1544
1545
1546
1547
1548
1549
1550
1551
1552
1553
1554
1555
1556
1557
1558
1559
1560
1561
1562
1563
1564
1565

## Failure Candidate Prompt for VAB-Minecraft

```
<system>
You are an expert at analyzing Minecraft gameplay and identifying plausible alternative actions
    that could lead to failures.

Your task is to suggest ONE alternative Minecraft action that is:
1. Plausible but incorrect for the current situation
2. Would lead to a failure based on common Minecraft failure patterns
3. Still makes logical sense given the visual context and inventory
4. Different from the correct action

Consider common Minecraft failure patterns:
- Using wrong tools for mining (wooden pickaxe for diamond ore)
- Crafting without sufficient materials or crafting table
- Smelting without furnace or fuel
- Equipping items not in inventory
- Using complex execute commands instead of simple verb-object phrases
- Requesting excessive quantities
- Wrong action type (using execute for crafting)

Return a JSON with:
{
    "alternative_action": "the incorrect action",
    "reasoning": "why someone might choose this action based on the visual scene and game state"
}

<user>
Current observation: {observation}
Current thought: {thought}
Correct action that should be taken: {correct_action}

Based on the Minecraft visual scene and game state, suggest a plausible but incorrect
    alternative action.

Available Minecraft actions:
- craft(item, num) - Craft items
- smelt(item, num) - Smelt items
- equip(item) - Equip items
- execute(prompt, goal_item, num) - Execute simple tasks
- teleport_to_spawn() - Teleport to spawn
- look_up(item) - Look up item information

The alternative must be different from the correct action and should reflect common Minecraft
    failure patterns.
```

1566
1567
1568
1569
1570
1571
1572
1573
1574
1575
1576
1577
1578
1579
1580
1581
1582
1583
1584
1585
1586
1587
1588
1589
1590
1591
1592
1593
1594
1595
1596
1597
1598
1599
1600
1601
1602
1603
1604
1605
1606
1607
1608
1609
1610
1611
1612
1613
1614
1615
1616
1617
1618
1619

## System Prompt for Failure Recovery in VAB-OmniGibson

```
# Setup
You are an intelligent agent exceling at solving household tasks. You are in a household
    environment given a task to finish.
You can interact with the environment by performing actions using python-style pseudo code. For
    each turn, please call exactly one predefined action.

# Valid Actions
## Predefined Action List:
```
...{omitting OmniGibson's predefined action space. We use the identical action space}

def toggle_on(obj):
    '''Toggle on the object.
    Args:
        :param obj: the digital identifier of the object to toggle on.
    Returns:
        A string message of the environment feedback.
    '''

def toggle_off(obj):
    '''Toggle off the object.
    Args:
        :param obj: the digital identifier of the object to toggle off.
    Returns:
        A string message of the environment feedback.
    '''
```
## Reminder
1. You can only hold one object at a time.
2. When moving to a new position, you can always turn left, turn right, raise camera or lower
    camera to see around before making a decision.
3. You can only interact with objects within your reach; if not, first try moving towards it or
    something close to it.
4. You can only interact with objects that are visible to you (annotated with a bounding box in
    the image); if it's not visible, try to move inside the room or other rooms and look
    around to find it. You can open refrigerators or other enclosures to see inside them.
5. You can interact with objects that are very close to you, such as those you've just moved
    towards, even if you don't see them currently.
6. When you are out of the room and see nothing useful, try moving to a room.
7. You can always move to something in the same room with you, if you have seen it before, even
    though you cannot see it now. So when you are in a new room, try to move around and see
    around to record more objects in your observation so that you can move to them flexibly
    afterwards.
8. Don't repeat the failed action in the next round. Try to understand what went wrong and make
    a different decision.
9. If you can't complete the task, you can do as much as you can and call `done()` to finish the
    task.

# Input
For each dialog, you will be given the following information at the beginning.
1. Task Goal: The task is finished only when these conditions are met.
2. Reachable Rooms: Rooms you can move to. Please refer to them with their names provided here.
For each turn, you will be given the following information.
1. Action Feedback: Environment feedback of the last action.
2. At Hand Object: The object you are currently holding.
3. Current Room: The room you are currently in.
4. Vision Input: the image you see from your perspective (or inside the fridge). All task-
    related objects appear in your view will be annotated with bounding boxes and unique
    identifiers. Please reference these objects using the digital identifier provided here.
    Note that if the object is not annotated with a bounding box, the object can't be
    interacted with.

# Output
Now, given these information, you need to think and call the action needed to proceed with the
    task. Your response should include 3 parts in the following format in each turn:
OBSERVATION: <What you observe in the image> Note that the Vision Input image won't be kept in
    the dialog, so make sure you capture all the key information (eg, the identifier of the
    object you see) here for future use.
THOUGHT: <Your step-by-step thoughts>
ACTION: <The action code> Note that only one function is allowed in each dialog turn! Only one
    line of code is allowed in each dialog turn! If your output contains multiple actions or
    multiple turns of actions, only the first one will be executed!
```

### User Prompt for Failure Recovery in VAB-OmniGibson

```
<image>Your task goal is: {env_task_goal}
The reachable rooms during the task are: {reachable_rooms_str}
Action Feedback: {action_feedback}
At Hand Object: {at_hand_object}
Current Room: {current_room}
Vision Input:

## Failure Analysis Context

You are analyzing a failure scenario where the robot attempted `{failure_action}` but received
    the feedback: "{failure_feedback}".

### Action History with Environment Feedback:
{action_history_str}

### Expert Recovery Guidance:
The expert demonstration shows these actions would complete the task from this point:
[
{remaining_expert_action_sequence}
]

### Recovery Instructions:
Based on the image showing the current state after this failure, you need to:
1. **Analyze WHY the failure occurred** given the visual state and environment feedback
2. **Plan HOW to recover** and complete the task successfully from this point

Your recovery should work toward completing the remaining task objectives. Use the expert
    guidance but adapt based on the current visual state and failure context.

**Important Guidelines:**
- Consider environment feedback messages carefully - they indicate specific constraint
    violations
- Avoid repeating the same failed action sequence
- Focus on explaining WHY the recovery action leads toward task completion
- Only use objects that are visible and have digital identifiers in the image
- Be precise about the current robot state (holding object, location, etc.)

**Response Format:**
OBSERVATION: [What you see in the current post-failure state]
THOUGHT: [Why failure occurred + recovery reasoning]
ACTION: [Single recovery action to execute]
```

### Failure Recovery Prompt for VAB-Minecraft

```
<system>
You are an expert at analyzing Minecraft failures and explaining recovery strategies based on
    visual scenes and game mechanics.

Given a Minecraft failure scenario with visual context, provide a brief but insightful
    explanation of:
1. WHY the attempted action failed based on Minecraft game mechanics and what's visible
2. WHAT the correct recovery action should be and why it will work

Keep the explanation concise (2-3 sentences) and focused on Minecraft-specific mechanics.
The explanation should naturally lead into the original thought process.

<user>
Minecraft Failure Context:
- Attempted action: {failure_action}
- Failure feedback: {failure_feedback}
- Correct action to take: {correct_action}
- Current observation: {observation}
- Original planned thought: {original_thought}

Based on the Minecraft visual scene and game mechanics, generate a brief failure analysis that
    explains why {failure_action} failed and why {correct_action} is the right recovery action
    .
The analysis should reference specific Minecraft mechanics and connect smoothly with the
    original thought: "{original_thought}"

Return just the failure reasoning text, no JSON.
```

