# OpenReview forum: "INFUSER: Injecting Synthetic Failures for Self-Correcting Embodied Agents"
_ICLR.cc/2026/Conference — ICLR 2026 Conference Withdrawn Submission_

### Official Review · Reviewer_DoXq · 2025-10-17

**Soundness:** 2
**Presentation:** 3
**Contribution:** 2
**Rating:** 2
**Confidence:** 4

**Summary:**

This paper introduces INFUSER, a framework designed to enhance embodied agents through synthetic failure recovery data. The synthetic data are generated by injecting suboptimal actions into ground-truth trajectories. By fine-tuning the model with both the synthetic failure recovery data and the original successful trajectories, the framework could improve agent performance. Experiments conducted on EB-ALFRED, EB-Habitat, VAB-OmniGibson, and VAB-Minecraft demonstrate consistent performance gains.

**Strengths:**

1.	Failure recovery is an important capability that an embodied agent should possess. This paper presents a useful framework for generating such data and enhancing the performance of small-scale models.

2.	The writing is clear and easy to follow.

3.	The experiments cover multiple benchmarks, demonstrating the generality of the approach.

**Weaknesses:**

1.	The proposed method appears to primarily target small models with limited performance through domain-specific fine-tuning. It would be beneficial to demonstrate how failure recovery contributes when the base failure rate is relatively low.

2.	The method relies on domain-specific fine-tuning, which limits its applicability to strong proprietary models. Moreover, this setting makes the comparison with zero-shot models somewhat unfair.

3.	As shown in Table 2, the performance gain on EB-Habitat from using failure recovery data is quite limited compared to the large improvements observed on EB-ALFRED. It would be helpful to discuss the reason behind this discrepancy and clarify the effective scope of failure recovery data. Additionally, including an ablation study (similar to Table 4) would strengthen the analysis of the effectiveness of failure recovery trajectories.

4.	(Minor) Figure 3 contains noticeable formatting errors.

**Questions:**

See weaknesses.

---

### Official Review · Reviewer_sWCg · 2025-10-25

**Soundness:** 2
**Presentation:** 2
**Contribution:** 2
**Rating:** 2
**Confidence:** 3

**Summary:**

This paper introduces INFUSER, a method for training failure-tolerant embodied agents by augmenting expert demonstrations with synthetically generated failure-recovery trajectories. The key insight is using an LLM to inject plausible failure actions into expert trajectories and generate corresponding recovery reasoning. The authors then fine-tune VLMs on this augmented dataset using teacher-forcing on expert recovery actions. Evaluated on EmbodiedBench and VisualAgentBench, INFUSER improves success rates and achieves the highest recovery rate.

**Strengths:**

1. Addresses a critical problem: Trains agents to recover from failures rather than just avoid them, which is essential for real-world deployment.

2. Comprehensive evaluation: Validates the approach across multiple benchmarks (EB-ALFRED, EB-Habitat, VAB-OmniGibson, VAB-Minecraft) with detailed error persistence analysis.

**Weaknesses:**

1. Unclear environment interaction during data generation: Equation (3) describes o_t_fail​​ as the observation before generating
a_t_fail​​, while the text below Equation (4) refers to it as "resulting observation." If the latter is correct, does it mean o_t_fail​+1​? This raises the question of whether environment interaction is required during dataset construction, which is not explicitly stated.

2. Lack of motivation for training scheme: The paper does not explain why teacher-forcing with expert recovery actions is necessary. Why not simply train the agent to return to previous states? Comparisons with simpler alternatives and in-depth justification are missing, making the experimental section read like a list of results rather than providing insights.

3. Insufficient baseline comparisons: The performance gains are unsurprising given the custom dataset. If environment interaction is needed for data generation (see weakness 1), why not directly fine-tune VLMs in the environment? More importantly, various dataset construction methods are possible (see weakness 2), and ablations comparing these alternatives are critically needed.

4. (Minor) Figure 3 caption formatting issue: The caption placement appears incorrect.

**Questions:**

Please see weaknesses above.

---

### Official Review · Reviewer_hEuz · 2025-10-31

**Soundness:** 3
**Presentation:** 2
**Contribution:** 2
**Rating:** 6
**Confidence:** 4

**Summary:**

This paper introduces INFUSER, a framework for training embodied agents to recover from failures by augmenting expert demonstrations with synthetically generated failure-and-recovery trajectories. The core idea is to inject suboptimal actions into ground-truth paths and train agents to recognize and correct these failures. The approach achieves significant improvements on EmbodiedBench and VisualAgentBench benchmarks.

**Strengths:**

1.The paper addresses a critical limitation in current embodied agent training - the lack of failure recovery capabilities when agents are trained exclusively on successful demonstrations.
2.The synthetic data generation pipeline requires no additional human annotation, making it scalable and practical compared to collecting expert failure demonstrations.
3.Substantial improvements on EB-ALFRED and EB-Habitat
4.The paper includes thorough ablations, error persistence analysis, and qualitative examples that provide insights into what the model learns.

**Weaknesses:**

1.Most tasks are relatively short-horizon. How does INFUSER perform on longer tasks with cascading failures?
2.Does training with synthetic failures on one task set help with entirely new tasks?
3.The paper primarily compares against supervised fine-tuning baselines. Missing comparisons include: Self-correction methods: Recent work on self-reflective agents (e.g., ReAct, Reflexion) - how do they compare? Human-in-the-loop: What if humans provided actual failure recovery demonstrations for the same cost as running GPT-4o?

**Questions:**

1.How does performance degrade when synthetic failures don't match real failure distributions?
2.Can the method work with less capable LLMs than GPT-4o for failure generation?

---

### Official Review · Reviewer_B2Q6 · 2025-11-01

**Soundness:** 2
**Presentation:** 2
**Contribution:** 2
**Rating:** 2
**Confidence:** 3

**Summary:**

This paper proposes INFUSER, a method for training failure-tolerant embodied agents by augmenting expert demonstrations with synthetically generated failure-recovery trajectories. The approach uses GPT-4o to inject plausible suboptimal actions into ground-truth trajectories and generate corresponding recovery reasoning. The authors evaluate their method on EmbodiedBench (EB-ALFRED and EB-Habitat) and VisualAgentBench (VAB-OmniGibson and VAB-Minecraft), showing improvements in task success rates and recovery capabilities compared to models trained only on successful demonstrations.

**Strengths:**

Practical Problem: The paper addresses a real limitation in current embodied AI systems - the inability to recover from errors when trained only on successful demonstrations.

**Weaknesses:**

### Major Issues:

1. **Limited Novelty**: The core idea of using synthetic failures for robust policy learning is not new. Similar concepts have been explored in:
   - **DAgger framework** (Ross et al., 2011 - already cited) introduced the idea of learning from both expert and agent's own mistakes
   - **Adversarial imitation learning** (Ho & Ermon, NeurIPS 2016, "Generative Adversarial Imitation Learning") explored learning robust policies through adversarial training
   - **Learning to generalize across tasks** (Mandlekar et al., RSS 2020, "Learning to Generalize Across Long-Horizon Tasks from Human Demonstrations") studied recovery from distribution shift in long-horizon tasks

   The paper's contribution is primarily an engineering application of existing ideas (synthetic data augmentation + LLM-generated reasoning) rather than a fundamental methodological advance.

2. **Problematic Experimental Design**:

   a) **Distribution Bias**: All synthetic data is generated using GPT-4o, creating a strong bias. The model essentially learns to mimic GPT-4o's reasoning patterns rather than developing independent recovery strategies. This raises the question: Is INFUSER learning general recovery principles or just learning to pattern-match GPT-4o outputs?

   b) **Train-Test Mismatch**: Critical issue with Equations 7-8. During training, the expert recovery action a*_{t_fail} is provided as input, allowing the model to focus on generating reasoning. However, at test time, the model must generate both reasoning AND actions without expert guidance. This is a severe distribution shift that is not adequately addressed or analyzed.

   c) **Unfair Baseline Comparison**: In Table 4, the baseline "Qwen2.5-VL-7B*" uses 6K successful trajectories while INFUSER uses 5K successful + 1K failure trajectories. The improvement from 18.3% to 33.0% could result from:
      - Better data diversity rather than failure-recovery learning
      - The different reasoning annotations
      - Simply having different training data composition

   A fair comparison requires: (1) Same total data size, (2) Same annotation quality, (3) Controlled variables.

3. **Missing Critical Ablations and Analysis**:

   a) **No comparison with data diversity baselines**: What if you simply add 1K more diverse successful demonstrations instead of failure trajectories? The paper doesn't prove that failures specifically help vs. general data diversity.

   b) **No analysis of synthetic failure quality**:
      - How realistic are GPT-4o generated failures?
      - Do they cover actual failure modes in deployment?
      - What's the distribution of failure types?

   c) **No multi-seed evaluation**: All results appear to be from single runs, making it impossible to assess statistical significance.

   d) **Missing ablation on failure types**: Which types of failures (planning vs. reasoning errors) contribute most to improvement?

   e) **No analysis of failure injection strategy**: Why uniform sampling? Would importance sampling or curriculum learning be better?

4. **Questionable Evaluation Methodology**:

   a) **Limited metrics**: Success rate alone doesn't distinguish between:
      - Models that avoid errors through better planning
      - Models that recover from errors through corrective actions

   b) **Arbitrary recovery definition**: "Recovery rate (errors corrected within 2 steps)" - Why 2 steps? How sensitive are conclusions to this choice?

   c) **No failure case analysis**: The paper doesn't discuss when INFUSER fails or what types of failures cannot be recovered from.

   d) **Missing computational cost analysis**: What's the overhead of generating synthetic data? What's the inference time comparison?

   e) **Statistical significance**: No error bars, confidence intervals, or significance tests reported.



5. **Limited Scope and Generalization**:

   a) **Simulation only**: All experiments in simulated environments with:
      - Perfect state information
      - Deterministic physics
      - No sensor noise
      - No real-world uncertainties

   b) **No sim-to-real discussion**: How would this approach transfer to real robots where:
      - Failures are more diverse and unpredictable
      - Recovery requires physical robustness
      - Computational resources are limited

   c) **Simple tasks**: Most tasks are relatively short-horizon (< 20 steps) household tasks. Scalability to complex, long-horizon tasks is unclear.

### Minor Issues:

6. **Presentation and Clarity**:

   a) **Excessive length**: Over 1600 lines with substantial redundancy. The appendix contains 30+ pages with overly detailed prompts that could be summarized.

   b) **Poor figure quality**: Figures 5-6 in appendix show similar scenarios without adding insights. Figure 4 comparison is too small to read clearly.

   c) **Inconsistent terminology**: "Failure-recovery trajectories", "failure-tolerant data", "synthetic failures" used interchangeably.

7. **Theoretical Gaps**:

   a) **No formal analysis**: Why should training on synthetic failures help with real failures? No theoretical justification or sample complexity analysis.

   b) **No failure taxonomy validation**: The failure taxonomy in Appendix C is presented without validation - do these categories cover real failure modes?

   c) **Missing generalization bounds**: Under what conditions does learning from synthetic failures generalize to real failures?

8. **Related Work Limitations**:

   a) **Incomplete coverage**: Missing important work on:
      - **Robot learning from demonstration surveys** (Argall et al., Robotics and Autonomous Systems 2009)
      - **Imitation learning theory** - recent advances in understanding distribution shift
      - **Robustness in embodied AI** - systematic studies on failure modes

   b) **Superficial comparison**: The related work doesn't clearly position the contribution relative to existing failure-aware learning methods. For example, how does this differ from ReWiND (Zhang et al., 2025 - cited) beyond using LLM-generated reasoning?

9. **Experimental Issues**:

    a) **Cherry-picked examples**: Figure 1 and Figure 4 show success cases. Where are the failure cases of INFUSER?

    b) **Inconsistent improvements**: Performance varies wildly across task categories (Table 1: from 2%→16% on "Long" to 8%→68% on "Base"). No explanation for this variance.

    c) **Comparison with wrong baselines**: Why not compare with more recent embodied AI methods from 2024-2025? The field is moving rapidly.

    d) **VAB results are marginal**: Table 3 shows only 3.3% improvement on VAB-OmniGibson (5.5%→8.8%) and 3.0% on VAB-Minecraft (28.4%→31.4%). These could be within noise levels without significance testing.


- Ho, J., & Ermon, S. (2016). Generative adversarial imitation learning. *NeurIPS*.
- Argall, B. D., Chernova, S., Veloso, M., & Browning, B. (2009). A survey of robot learning from demonstration. *Robotics and Autonomous Systems*, 57(5), 469-483.
- Mandlekar, A., Xu, D., Martín-Martín, R., Savarese, S., & Fei-Fei, L. (2020). Learning to generalize across long-horizon tasks from human demonstrations. *RSS*.

**Questions:**

Data Quality: How do you ensure the synthetic failures are realistic and cover the actual failure modes that occur during deployment? Can you provide statistics on the diversity of generated failures?

Generalization: How does the method perform on out-of-distribution failures that differ from the synthetic failures generated by GPT-4o?
Ablation Studies:

What happens if you train without the expert action guidance (a*_t) from the beginning, rather than just at test time?

How sensitive is performance to the failure injection rate ρ for different task complexities?
What if you use a weaker LLM (e.g., GPT-3.5) for synthetic data generation?


Real-world Applicability: Have you conducted any real robot experiments? What are the main challenges in transferring this approach to real-world scenarios?

Computational Cost: What is the computational cost of generating the synthetic data? How does the training time compare between baseline and INFUSER?

Failure Analysis: Can you provide examples where INFUSER fails to recover? What are the limitations of this approach?

---

### Note · Authors · 2025-11-14

I have read and agree with the venue's withdrawal policy on behalf of myself and my co-authors.